# Therapeutic Effects of WT1 Silencing via Respiratory Administration of Neutral DOPC Liposomal-siRNA in a Lung Metastasis Melanoma Murine Model

**DOI:** 10.3390/ncrna9020021

**Published:** 2023-03-22

**Authors:** Martin R. Ramos-Gonzalez, Eduardo Vazquez-Garza, Gerardo Garcia-Rivas, Cristian Rodriguez-Aguayo, Arturo Chavez-Reyes

**Affiliations:** 1Department of Genetic Therapy, Monterrey Unit, Center for Research and Advanced Studies of the National Polytechnic Institute, Monterrey 66600, Nuevo León, Mexico; mr.ramos@health.missouri.edu; 2Cátedra de Cardiología Y Medicina Vascular, Escuela de Medicina, Tecnologico de Monterrey, Monterrey 64849, Nuevo León, Mexico; 3Department of Experimental Therapeutics, University of Texas MD Anderson Cancer Center, Houston, TX 77054, USA; 4Basic Sciences Unit, Medical School, Universidad Finis Terrae, Santiago de Chile 7501015, Chile

**Keywords:** melanoma, lung cancer, metastases, siRNA, liposomes

## Abstract

The lungs represent a frequent target for metastatic melanoma as they offer a high-oxygen environment for tumor development. The overexpression of the WT1 protein has been associated with the occurrence of melanoma. In this study, we evaluated the effects of silencing the WT1 protein by siRNA in both in vitro in the B16F10 melanoma cell line and in vivo in a murine model of lung metastatic melanoma. We did this by implementing a novel respiratory delivery strategy of a neutral DOPC liposomal-siRNA system (L-siRNA). In vitro studies showed an effective silencing of the WT1 protein in the siRNAs’ WT1-treated cells when compared with controls, resulting in a loss of the cell’s viability and proliferation by inducing G1 arrest, the inhibition of the migration and invasion capacities of the cells, as well as the induction of apoptosis. In vivo, the respiratory administration of L-WT1 siRNA showed an efficient biodistribution on the lungs. After two weeks of treatment, the silencing of the WT1 protein resulted in an important antitumor activity that reduced the tumor weight. In the survival study, L-WT1 treatment could significantly delay the death of the animals. This work demonstrates the efficacy of the L-siRNA respiratory administration as a novel therapy to reduce pulmonary tumors and to increase survivability by silencing specific cancer oncogenes as WT1.

## 1. Introduction

Cancer is one of the main causes of death around the world, with the most significant being lung, liver, stomach, colon, and breast cancers. As a major concern, the dissemination of metastases is responsible for more than 90% of deaths by cancer. Tumor cells spread by lymphatic and blood vessels from the primary site of origin to distant organs such as the lungs, brain, bones, and liver [1]. The lungs are the second most common site of metastases. This is because its location supports the high demand of nutrients tumor cells require, as well as because of its extensive vascular surface, high levels of oxygenation, and having enough space to grow [2].

Melanoma has a high mortality rate, and is responsible for 80% of the deaths related to skin cancer [3]. Its high malignancy is due to its ability to disseminate metastases to distant tissues, as well as its ability to adapt to different microenvironments in various organs [4]. The limited number of options to treat melanoma brings the necessity to develop new therapeutic strategies that can target the complexity of the molecular pathways implied in the development and maintenance of cancer cells. This can also be applied to distant sites of metastasis, allowing for the reduction in chemotherapy resistance [5]. Recently, the identification of different molecular biomarkers can predict or modulate the response of a specific type of cancer to treatment. This can also be used as targets for directed therapy that can directly kill the tumor or sensibilize it to other conventional treatments [6]. In this category of targeted biomarkers is the Wilms tumor 1 gene (*WT1*). This gene was first identified in 1990 from a pediatric nephroblastoma that was initially described by Doctor Max Wilms in 1899 [7]. *WT1* codifies a transcription factor for the demonstrated gene, showing that the absence of this gene is incompatible with life [8]. There are at least 24 known isoforms of the WT1 protein. The most important ones being threonine and serine (+KTS and −KTS, respectively) in exon 9 [9,10].

*WT1* has been described as an oncogene and its high expression has been detected in several different types of neoplasms, such as breast cancer [11], mesothelioma [12], and leukemias. Leukemias in particular have a poor prognosis, with a less than 12% chance of survival over 3 years [13]. It also has a high expression in several melanoma cell lines but is not found in normal melanocytes [14]. There is a great diversity of genes targeted by *WT1*, and these genes have a role in the process of malignant transformation. These regulatory genes include modulators of proliferation, cell cycle progression, apoptosis, growth factors, and their receptors [15].

A novel strategy that has been widely used as a regulatory mechanism is silencing the expression of a desired gene. For example, the RNA interference (RNAi) works by the degradation of specific mRNA [16]. When a double strain RNA enters the cell, as small interfering RNAs (siRNAs), it activates the siRNA-mediated cell silencing process mediated by Dicer and produces a cut in a targeted mRNA. This results in silencing by not allowing the protein to be produced [17]. Since the year 2000, the use of siRNAs as a silencing tool has been more widely employed as a treatment for hereditary diseases, antiviral therapy (HIV, HBV, HCV), and even as a vaccine for certain types of tumors [18].

There are antecedents of the use of RNAi as an experimental treatment to target lung tumors. The work of Zamora-Avila et al. [19] focused on the administration of a plasmid codifying for short hairpin RNA (shRNA) against *WT1* using polyethylenimine (PEI) as a vector in a therapy for lung melanoma metastases. In this work, we focus on the use of neutral DOPC-liposomes as siRNA carriers, as they offer certain advantages such as their low cost, their simpler production method, and their phospholipidic nature makes them ideal to interact with cells [20]. We utilize neutral liposomes of DOPC as they offer additional benefits such as a low toxicity, a high encapsulation efficiency of siRNA, and not being immunogenic [21].

## 2. Results

### 2.1. Silencing of WT1 Reduces Cellular Malignancy Phenotype

Our first objective was to evaluate the efficacy of our WT1 siRNA to silence the expression of the WT1 protein in the B16F10 melanoma cell line. After 72 h of treatment with siRNA, we recovered whole protein lysate from the cells and detected the WT1 protein via a western blot. We observed a significant silencing of the WT1 protein expression, decreasing by 86% when compared to the untreated control (Figure 1A). This silencing effect is specific of the sequence as the control siRNA had no significant reduction on the WT1 protein, even when the same transfection agent and molecular entity was used (Control siRNA).

After confirming that we could achieve a potent silencing effect of *WT1* with our siRNA sequence, we sought to analyze if this protein was involved in conferring a metastatic phenotype to the melanoma cells by changing its motility properties. For this purpose, we performed a migration assay with a FBS gradient through the transwell membrane. This process requires an active movement of the cells that could be measured by using DAPI fluorescent nuclear staining on the cells that crossed the membrane. After 48 h, we observed that the WT1 siRNA-treated group showed an 88% reduction in migration against the control (Figure 1B). Even more impressive, in the invasion assay using transwells coated with Matrigel, the treatment with WT1 siRNA after 72 h completely restrained the cell’s invasive capacity (Figure 1C).

### 2.2. WT1 Silencing Induces Arrest of Cell Cycle and Reduces Cell’s Survival

To elucidate the importance of *WT1* in the cell cycle progressions, we incubated the melanoma cells with WT1 siRNA for 72 h. After the elapsed time, we applied a propidium iodide staining and then evaluated the stage of the cell cycle using a flow cytometer. In the group treated with WT1 siRNA, we observed an arrest on G1 by around 11%, and a reduction in the G2/M phase by 63% when compared to the untreated group. No significant changes were observed between the two control groups (Figure 2A). This result is accompanied by a sharp decline in viability of 40% in the melanoma cells treated with WT1 siRNA after 72 h (Figure 2B). It is worth mentioning that the use of the transfection agent with the Control siRNA had a small effect of 8% over the viability.

Another mechanism that *WT1* is involved in is cell survival. To evaluate the importance of *WT1* in the process of apoptosis we performed a TUNEL assay to determine if the silencing of *WT1* could induce the activation of apoptosis. We seeded melanoma cells over a cover slide and incubated with WT1 siRNA for 72 h, then processed the samples for TUNEL fluorescent assay. We found the silencing of *WT1* activates apoptosis, detecting positive nuclei in as much as 10% of the total cells (Figure 2C). These results show the effect over viability has a multifactorial explanation, as this reduction in the cell’s number can be explained by the contribution of the cell’s cycle arrest and apoptotic induction. With this we highlight the importance of *WT1*′s role in cell division, survival, and the evasion of apoptosis.

### 2.3. Biodistribution Study

Once we confirmed the efficacy of silencing *WT1* as a mechanism to diminish the malignancy phenotype in vitro, we wanted to further expand the application of this siRNA therapy on an animal model bearing melanoma metastases on the lungs. For this purpose, we implemented a neutral DOPC liposomal system as a carrier of siRNA for in vivo biodistribution that we administered using a nebulized inhalatory formulation that could target lungs as a primary source of delivery.

For the biodistribution study, C57BL/6 mice were inoculated with a melanoma B16F10 cell via a tail vein to produce lung metastases. After 2 weeks of tumor development, the animals were placed in a ventilatory system so the animals could breathe the Alexa 488-labeled liposomal siRNA formulation (L-siRNA). The liposomes were allowed to be distributed through the animal’s body, and after 1 h we obtained different organs to evaluate the presence of an Alexa 488 fluorescence signal. Figure 3A shows the control group without any fluorescence marker in both the brightfield (left pictures) and 488 fluorescent (right pictures) channels. Figure 3B shows tissues recovered from animals which were administered L-Control siRNA-Alexa 488/We observed a high concentration of Alexa 488 on the lungs, indicative of the presence of L-siRNA on the target organ. We can also detect the presence of the Alexa 488 signal in depuration organs such as the liver, kidney, and spleen, though not as intense as in the lung. These results confirm the delivery of our L-siRNA formulation through the respiratory system.

### 2.4. Silencing of WT1 by L-siRNA Reduces the Size of Lung Metastases

After demonstrating the ability to deliver L-siRNA to the lungs, we proceeded to test the potential of this respiratory administration for the delivery of L-WT1 siRNAs as an antitumoral therapy for melanoma metastases in vivo. Mice bearing lung melanoma metastases were treated with respiratory L-siRNA twice per week for 3 weeks. At the end of the treatment, we obtained the lungs to evaluate the extension of the melanoma metastases. The group treated with L-WT1 siRNA showed a smaller number of metastatic foci, as well as an important reduction in the size and extension of these metastases (Figure 4A). Since these metastatic nodules are present in the whole depth of the lung tissue, as a quantitative parameter for tumoral extension we weighted the whole lungs that were carrying the melanoma metastases, observing a total weight reduction of 41% in the L-WT1 siRNA group (Figure 4B). From the lungs obtained we extracted the total protein to evaluate the capacity of our L-siRNA to silence the WT1 protein. We detected via a WB a reduction of 59% in the detectable WT1 protein in the L-WT1 siRNA treatment group (Figure 4C,D). These findings confirm the efficacy of silencing *WT1* as a therapy to limit the growth rate of metastases allocated in the lungs.

### 2.5. Survival Study

We confirmed the administration of L-WT1 siRNA achieved an important antitumoral effect after 3 weeks of treatment. Based on this we planned a survival study to assess the efficacy of silencing *WT1* for a longer period of time. The animals bearing lung melanoma metastases received the same treatment scheme of two doses per week of respiratory L-siRNA and their behavior was carefully followed until they reached endpoint criteria. The first animal deaths were observed at day 21 and 100% mortality was reached at day 27 in both control groups. However, animals in the L-WT1 siRNA group survived longer, with the first death observed at day 24 and full group mortality at day 34. This represents a full week extension on the survival of the L-WT1 siRNA group over the controls, demonstrating a 25% longer survival time (Figure 5A).

## 3. Discussion

In this work, we focused on testing the efficacy of a neutral DOPC-liposomal system for the delivery of siRNAs administered in a non-invasive way. This was performed by treating mice that were induced with melanoma that ultimately metastasized to the lungs. The lungs are one of the organs most affected by metastatic cells, as it offers plenty of space to grow, high blood irrigation, and access to nutrients and oxygenation. All these characteristics make this organ an ideal target for tumors.

Modern medical approaches propose the use of novel therapies against cancer by identifying molecular biomarkers that allow tumoral cells to differentiate from healthy cells. By manipulating the expression of these biomarkers, we can make tumors more susceptible to traditional therapies, target tumoral cells so that they are easier to be identified and recognized by the immune system, reactivate mechanisms controlling the cellular cycle, or introduce apoptosis. In this instance, *WT1* is a promissory therapeutic target as it is overexpressed in many human cancers, including the transformation from normal melanocytes to melanoma [14,22,23].

To modulate the expression of these biomarkers as *WT1*, we utilize one of the most effective molecular tools, RNAi. This mechanism allows the silencing of a specific target gene by degradation of its mRNA. The high efficacy for silencing proteins by siRNA has been demonstrated in several studies as a novel cancer therapy. These studies focus on targeting different mechanisms of cancer malignancy, such as by limiting the process of angiogenesis [24,25], the activation of apoptosis [26], or by blocking viral proteins in virus-induced cancers [27]. However, in most cases, intratumoral injections or the use of naked siRNA cannot be translated into clinical practice. To guarantee an efficient delivery of the siRNAs to the target site, a carrier is required that can offer protection in the blood circulation against the effects of nucleases or renal depuration. Most carriers have disadvantages, such as immune system activation and tolerance when using viral capsids [28], the risk of genomic integration of the plasmids [29], toxicity derived via the use of electrostatic charges [19], or the cytotoxicity caused by the use of cationic liposomes [20]. To overcome these limitations, we utilized a system to encapsulate and deliver siRNA using neutral DOPC-liposomes. In blood circulation, neutral liposomes can pass through the capillary membranes because of their small size (less than 1 µm), distributing systemically and concentrating inside of tumors, taking advantage of the chaotic turbulence of the tumoral vasculature delivering the siRNA therapy in the affected site [21].

This work demonstrates the efficacy of siRNA to downregulate the overexpression of proteins that can act as oncogenes, reducing in tumoral cells its phenotype of malignancy. Similar results were obtained by Graziano et al., [30] by silencing with siRNA the protein WT1 in the MG-63 human osteosarcoma cell line. With the treatment, they observed a 44% decrease in cellular proliferation. Similarly, we observed a 49% inhibition of proliferation. When analyzing the effect of *WT1* silencing over the cell cycle, we observed an increase in G1 arrest, similar to what was reported by Kudoh et al. [31]. These findings correlate with Graziano et al. that the observed induction of cell cycle arrest by changes in the expression of proteins such as cyclin E, Cyclin D1, Retinoblastoma, p27, and p53 [32] proteins with an important role in G/S cell progression. He also found an increase in the ratio Bax/Bcl2, and procaspase 3 levels, apoptosis inductors after treatment. As a comparison, their findings required 10 to 100 times higher the concentration of siRNA than our work, but it might be related to their use of a different anti-WT1 siRNA sequence and a different cell line than the ones we used. We also evaluated the effect of WT1 siRNA over migration and invasion, obtaining very promising results with a decrease of migration of 88% and a complete inhibition of cellular invasion. Their results were obtained in a study of *WT1* silencing but with the use of microRNAs on lung cancer cells [33].

Being the lung the second most affected organ by tumoral metastases, we proposed a feasible therapy using L-siRNAs as a treatment of lung metastases. For this purpose, we designed a system to deliver encapsulated siRNA in neutral DOPC-liposomes, administered by respiratory nebulization. Our L-siRNA could be delivered to the lungs with simple administration, achieving the silencing of the *WT1* oncogene, and obtaining a significant therapeutic effect over the melanoma metastases on the lungs of our animal model. This modality of therapy results in a very comfortable non-invasive high levels of stress derived from frequent visits to the hospital and intravenous administrations.

One of the main contributions of the present study is the validation of a novel respiratory administration of neutral DOPC-liposomal siRNA therapy as a treatment for lung metastases. To corroborate the efficacy of our respiratory delivery system, and to extrapolate the obtained results from the in vitro experiments, we used a lung melanoma metastases mouse model to evaluate the therapeutic efficacy of our L-siRNA formulation. Our previous experience in lyophilizing L-siRNA [21] allowed us to produce multilamellar liposomes that have the capacity to penetrate the capillary’s layer of endothelial cells and because of their size (average 680 nm) they tend to accumulate in the tumor as its irregular vasculature creates turbulent blood flow that facilitates the liposomal extrusion to the tumoral tissue [20]. The approach in our in vivo model was to evaluate the biodistribution of our respiratory L-siRNA. We found a high concentration of the L-siRNA such in the liver and kidney, in animals with lung metastasis for 3 weeks. None of the treated mice showed any symptoms of respiratory distress during the administration of the nebulized therapy. After reaching the endpoint, L-WT1 siRNA mice showed smaller and fewer visible nodules. These results suggest that the efficacy of our respiratory system to silence oncogenes in the lungs had a potential therapeutic effect. It is worth noting that even when the results were favorable, the L-WT1 siRNA treatment could not completely eliminate the presence of metastases in the lungs, so the need to evaluate this treatment for longer periods of time. To understand the effects of the L-WT1 siRNA treatment in the long term, we performed a survival assay extending the treatment until the animals showed the presence of endpoint criteria, and then compared the survival between treatment groups. We observed that treatment with L-WT1siRNA improved the survival of the animals, and they stayed in better health conditions for a longer time, improving their life expectancy. Even though the animals could live longer with the treatment, still they developed melanoma metastases in the lung at a slower rate. We suggest that there are mechanisms that locally affect the efficiency of the L-siRNA distribution, such as fibrosis formation after tumoral cells invade the tissue, an inflammatory process that might be thickening the interstitial spaces blocking the liposomal diffusion to the cells, or the progressive diminish on tidal volume of the affected lungs caused by the increase in tumor size. The increase in tumor size limits the respiratory volume, and therefore decreases the absorbed dose in the lungs. The consequences of any of these scenarios is a restriction in the capture of L-siRNAs by the cells.

This work offers an expanded vision of the diverse mechanisms that participate in the process of malignancy of the melanoma cell line B16F10 that are dependent on *WT1*. We describe the effects of the silencing of this protein by quantifying its expression level, migration and invasion, viability, progression in the cell cycle, and apoptosis. We also propose a new administration system that is efficient, safe, non-invasive, comfortable, and of a low cost to deliver siRNA encapsulated in neutral liposomes. We further describe the biodistribution, advantages and limitations, as well as the therapeutic effects of this treatment in the short and long term in a model of metastatic lung cancer in mice. Looking ahead of our work, we propose that there is strong evidence pointing to the belief that the efficacy of our treatment can be widely improved by the use of concomitant therapies. It is feasible to think that silencing the oncogene of these tumoral cells makes them more sensitive to chemotherapy or radiotherapy and could imply the use of lower doses minorizing the appearance of adverse effects, greatly improving the quality of life of the patients and possibly achieving a complete remission of the tumors. Additionally, the application of respiratory L-siRNA is not limited to the treatment of lung cancer; it can also be effective to control the expression of inflammatory factors in pneumonia, slow the progression of hereditary diseases, and even prevent the spread of respiratory viruses by silencing important proteins involved in their replication.

## 4. Materials and Methods

### 4.1. Cell Line and Culture Conditions

The murine melanoma B16F10 cell line was acquired from the ATCC (Manassas, VA, USA), catalog CRL-6475. The cells were nurtured with Gibco’s DMEM/F12 medium (Thermo Fisher, Waltham, MA, USA, Cat. 11320033) supplemented with 10% fetal bovine serum (FBS) (Sigma-Aldrich, St. Louis, MO, USA, Cat. F0926-500ML) and maintained in a humidified incubator at 37 °C with 5% CO_2_. The number of passages did not exceed 30 after initial stock was recovered. The cell line was confirmed as mycoplasma negative through a PCR analysis using the Universal Mycoplasma Detection Kit (ATCC, Cat. 30-1012K).

### 4.2. siRNA and Transfection

siRNA sequences were commercially acquired from Sigma-Aldrich, including the *WT1*-silencing siRNA from the reported sequence (5′-AAGCUGUCCCACUUACAGAUGCdTdT-3′) and a universal negative control siRNA (Sigma-Aldrich, Cat. SIC001). For the transfection procedure, we prepared transfection complexes using the RNAiFect Transfection Reagent (Qiagen Sciences, Germantown, MD, USA, Cat. 301605) and siRNA (6:1, respectively) as according to the 3 groups formed: Untreated (0 µg), Negative Control (Control siRNA) (7 µg siRNA), and *WT1*-silencing (WT1 siRNA) (7 µg siRNA). In 6-well culture plates we seeded 1 × 10^5^ cells per well in a total volume of 2 mL of medium and let it attach to the bottom. The next day cells were treated with the transfection complexes and kept at 37 °C. According to the efficacy study reported by Landen et al., [21] cells were harvested at 72 h as this time point shows the highest silencing efficacy after siRNA administration.

### 4.3. Western Blot

Total protein was extracted from cultured cells after transfection using a NP-40 lysis buffer (150 mM NaCl, 50 mM Tris and 1% Triton X-100, pH 8.0) containing protease-inhibitor 1× (Promega, Fitchburg, WI, USA, Cat. G6521). A total of 50 µg of protein was loaded on an 8% SDS-PAGE gel; after electrophoresis, the resolved proteins were transferred to a PVDF membrane. Antibodies for WT1 (Santa Cruz Biotechnology, Paso Robles, CA, USA, Cat. sc-192) and β-actin (Sigma, Cat. A2228) detection were utilized as detailed by the manufacturer. β-actin was detected after membrane stripping and used as a loading control. As a chemiluminescent substrate, we used Pierce ECL for the WB (Thermo Fisher, Cat. 32209).

### 4.4. Migration and Invasion Assay

For the evaluation of the migration capacity of the cells, 24-well transwell inserts with 8-µm pores (Corning Life Science, Tewksbury, MA, USA, Cat. 3530097) were used, and for invasion assay a coat of Matrigel (Corning, Cat. 354248) at 1.2 mg/mL over the insert was applied. B16F10 cells were cultured in a serum starving condition using DMEM/F12 with 1% FBS for 24 h. After this time, cells were transfected corresponding to the treatment group. At the bottom of the transwell, 500 µL of DMEM/F12 with 10% FBS were added, and at the top of the insert, 2.5 × 10^4^ cells were seeded in 200 µL of DMEM/F12 with 1% FBS to produce a serum chemotaxis gradient. After 48 h the transwell inserts are collected for migration, and after 72 h for the invasion assay. All non-migrating cells were scrapped from the top of the insert and the remaining cells at the bottom were gently washed with PBS and fixed with 4% paraformaldehyde for 15 min. Transwell membranes were stained with nuclear dye DAPI (Vector Laboratories, Burlingame, CA, USA, Cat. H-1200) and mounted over a slide to allow its visualization and counting on a fluorescent microscope.

### 4.5. Cell Cycle Cytometry

Transfection and seeding of 1 × 10^5^ cells were performed in a 6-well culture plate with 2 mL of DMEM/F12 with 10% FBS. Treated cells were collected after 72 h and stained with ice-cold propidium iodide for 5 min before being analyzed via flow cytometry (BD Bioscience, San Jose, CA, USA, FACSCanto II). An unstained control was used to determine basal fluorescence. All data analyses were performed utilizing the ModFit LT v5 software.

### 4.6. Cell Viability

For the cell viability assay the AlamarBlue (Thermo Fisher, Cat. DAL1100) method was used. For each well (6 wells per group), 1 × 10^4^ cells were transfected and seeded in 96-well culture plates with 100 µL of DMEM/F12 with 10% FBS. After 72 h, plates were collected and read using an ELISA plate reader at a 570-nm wavelength. The experiment was conducted using a triplicate method.

### 4.7. Apoptosis TUNEL

A total number of 2.5 × 10^4^ cells were seeded over a round coverslip placed in 12-well plates with 1 mL of DMEM/F12 with 10% FBS. After 72 h, the coverslips were washed with PBS and stained with the kit DeadEnd Fluorometric TUNEL System (Promega, Cat. G3250) following the merchant’s instructions. Lastly, the coverslips were mounted over a slide using VectaShield with DAPI (Vector Laboratories, Cat. H-1300) and visualized in a fluorescent microscope.

### 4.8. Animal Model of Melanoma Metastasis

A pulmonary melanoma metastatic model was stablished on C57BL/6 mice via inoculation via the tail vein with 5 × 10^5^ cells of the syngeneic melanoma cell line B16F10. The cells were administered on 200 µL of physiological saline solution (NaCl 0.9%). This model develops visible tumoral nodules on the lungs between 7–15 days after I.V. administration.

### 4.9. Nebulization of Liposomal siRNA Treatment

To administer siRNA in vivo we followed the methodology previously described by Landen et al. [21] to prepare lyophilized liposomes that can be stored for long periods of time at −20 °C. The individual dose administered for every mouse was 10 µg of siRNA via resuspending on 500 µL of physiological saline solution. Each animal was placed in an individual nebulization capsule with limited movement, inhaling nebulized liposomal-siRNA (L-siRNA) for around 6 to 8 min, the total length of the treatment. The treatment was administered starting at the 3rd day after cell inoculation and keep at a rate of 2 doses per week for the total length of each experiment, being 3 weeks for the effect of *WT1* silencing evaluation, and a maximum of 8 weeks or until reaching endpoint criteria for the survival study.

### 4.10. Biodistribution

For this study, two groups of 3 mice each were inoculated with B16F10 melanoma cells via the vein of the tail and the metastases left to implant for 15 days. A control group received the L-siRNA control, and a second group received the L-siRNA control with the fluorescent marker Alexa fluor 488 (L-siRNA Alexa 488). Both groups were administered L-siRNA via nebulization as described before and left to rest for 1 h to allow the systemic distribution of the L-siRNA. After that time, the animals were euthanized and samples from multiple organs were recovered to be embedded in Tissue-Tek (VWR, Radnor, PA, USA, Cat. 25608). Frozen sections were processed using a Leica Cryostat (Mod. CM1100), and a fluorescent signal was observed in a confocal microscope (Leica Biosystems, Buffalo Grove, IL, USA, Mod. TCS SP5).

### 4.11. Survival Study and Animal Termination Criteria

To evaluate the long-term efficacy of our L-siRNA treatment, we implemented a survival study inducing melanoma lung metastases as detailed before with 10 animals assigned to each treatment group. The animals received nebulized therapy with L-siRNA twice per week, and their general condition and behavior was carefully followed daily to detect animals that started to show signs of physical affectation caused by the growth of the tumors in the lung. Endpoint criteria included respiratory distress, reduced movement, general weakness, and slow reaction times after a touch stimulus.

### 4.12. Tumors Recovery and WT1 Expression Analysis

Animals reaching endpoint criteria were euthanized in a CO_2_ chamber and a full set of organs were recovered and stored for further analysis. We paid special attention to the lung since it was the primary organ affected by metastases, and it was weighted and separated in different fragments for study. One of the analyses we performed in the tumor-bearing lungs was the detection of the WT1 protein via a western blot, and for that we recovered the whole-lung protein extract using RIPA Buffer (Thermo Fisher, Cat. J62524.AD).

### 4.13. Statistical Analysis

All analysis were performed using GraphPad Prism 9.0.2. On graphs, bars represent the mean ± SDs of at least 3 independent experiments. For the comparison of the multiple groups’ comparison, a one-way ANOVA was used, and for the survival study, Kaplan–Meier analysis was compared with Log-rank. A value of *p* < 0.05 was considered to be statistically significant.

## Figures and Tables

**Figure 1 ncrna-09-00021-f001:**
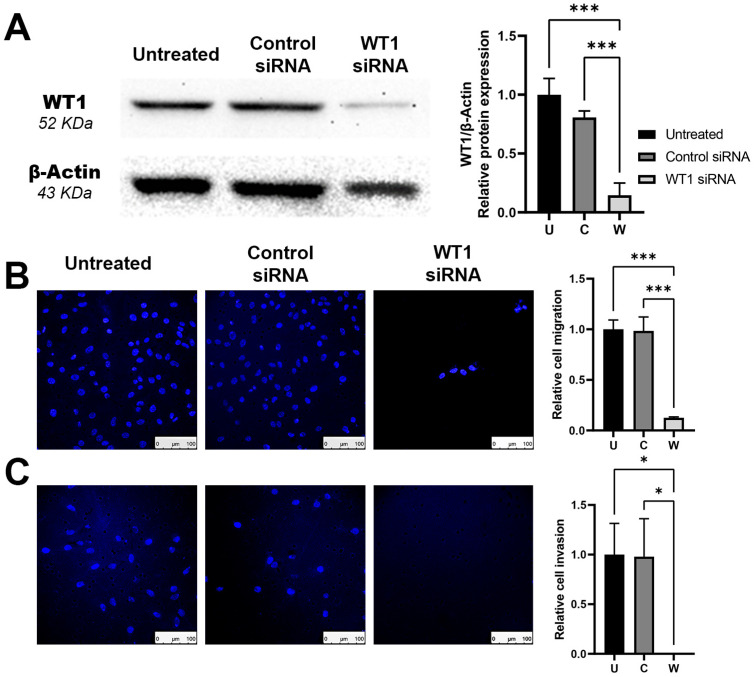
Effect of *WT1* silencing over protein expression and cellular motility on melanoma cells. (**A**) Western blot showing silencing of the WT1 protein and densitometry quantification for each group after 72 h of treatment. (**B**) Cell nuclei stained with DAPI after *WT1* silencing showing a reduced migration after 48 h and (**C**) a complete loss of invasion capacity through the Matrigel- transwell chamber after 72 h. Average number of nuclei per field was normalized to untreated group. Scale bars represent 100 µm. Groups U: Untreated, C: Control siRNA, W: WT1 siRNA. One-way ANOVA was used as statistical test for group’s comparison. * *p* < 0.05, *** *p* < 0.001.

**Figure 2 ncrna-09-00021-f002:**
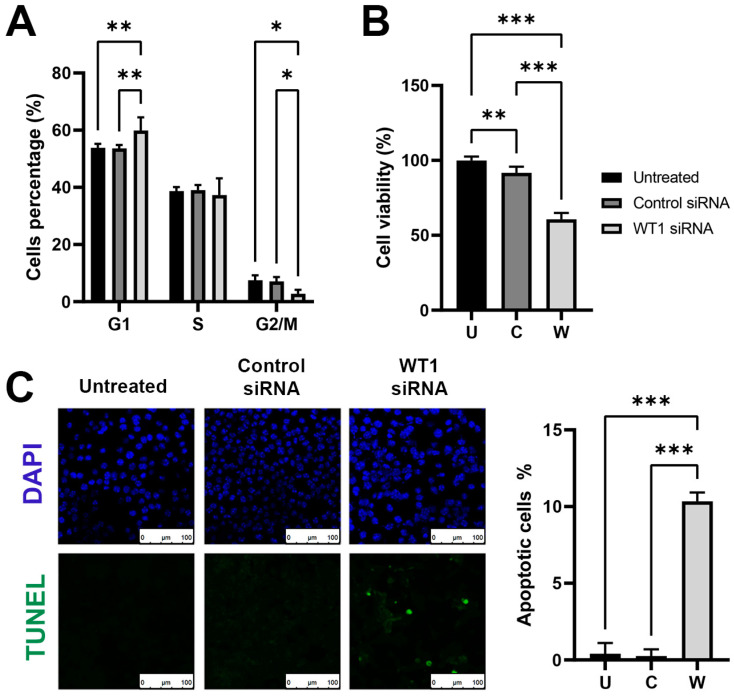
Induction of cell cycle arrest decreased viability and apoptosis activation after *WT1* silencing on melanoma cells. (**A**) *WT1* silencing induces G1 cell cycle arrest and decreased G2/M progression measured via PI cytometry. (**B**) AlamarBlue assay shows a reduced viability in the group treated with WT1 siRNA. (**C**) Apoptosis detection via a TUNEL assay after *WT1* silencing using fluorescent microscopy. The graph shows the percentage of TUNEL-positive nuclei per field compared to the untreated group. All experiments were performed after 72 h of treatment. Groups U: Untreated, C: Control siRNA, W: WT1 siRNA. Scale bars represent 100 µm. One-way ANOVA was used as statistical test for group’s comparison. * *p* < 0.05, ** *p* < 0.01, *** *p* < 0.001.

**Figure 3 ncrna-09-00021-f003:**
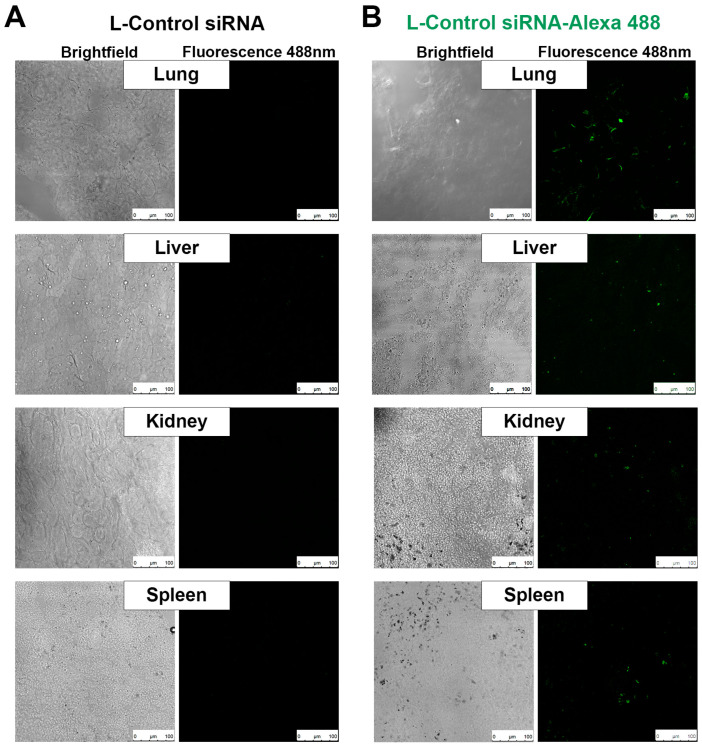
Respiratory delivery of liposomal siRNA biodistribution assay. Frozen sections of different tissues after 1 h treatment with (**A**) L-Control siRNA or (**B**) L-Control siRNA with fluorescent mark for Alexa 488 (in green). Comparison of brightfield and fluorescent images showing the lungs as the organ with the highest intensity of signal after the administration of respiratory L-siRNA. Scale bars represent 100 µm.

**Figure 4 ncrna-09-00021-f004:**
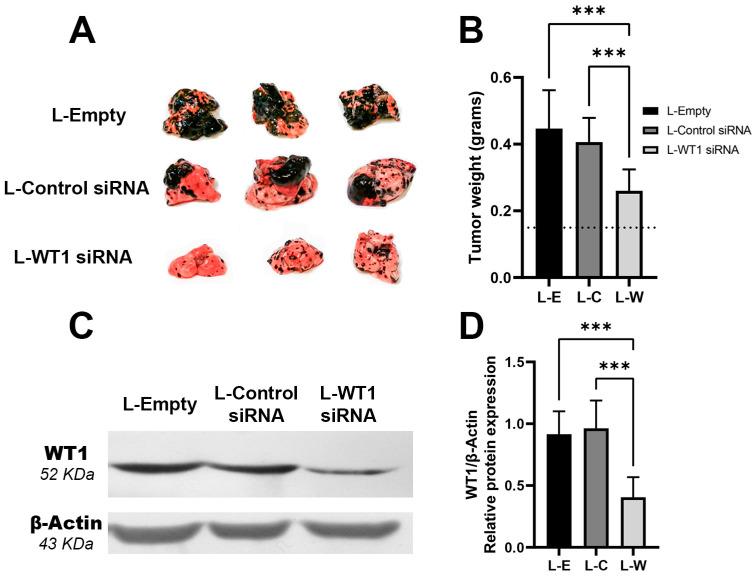
Respiratory administration of liposomal WT1 siRNA reduces the tumor size by decreasing WT1 protein expression in a lung metastatic melanoma model in mice. (**A**) L-WT1 siRNA administration reduces the size of the tumor and number of metastases in the lungs after 2 weeks of treatment. (**B**) There is a decrease in the weight of the lungs treated with L-WT1 siRNA when compared with control groups. Dot line represents average weight of a healthy tumor as a reference. (**C**) WT-1 protein expression is reduced in the lung’s melanoma metastases of the L-WT1 siRNA-treated group observed via a WB and (**D**) WT-1 quantification via densitometry of the bands. Groups L-E: L-Empty, L-C: L-Control siRNA, L-W: L-WT1 siRNA. One-way ANOVA was used as statistical test for group’s comparison. *** *p* < 0.001.

**Figure 5 ncrna-09-00021-f005:**
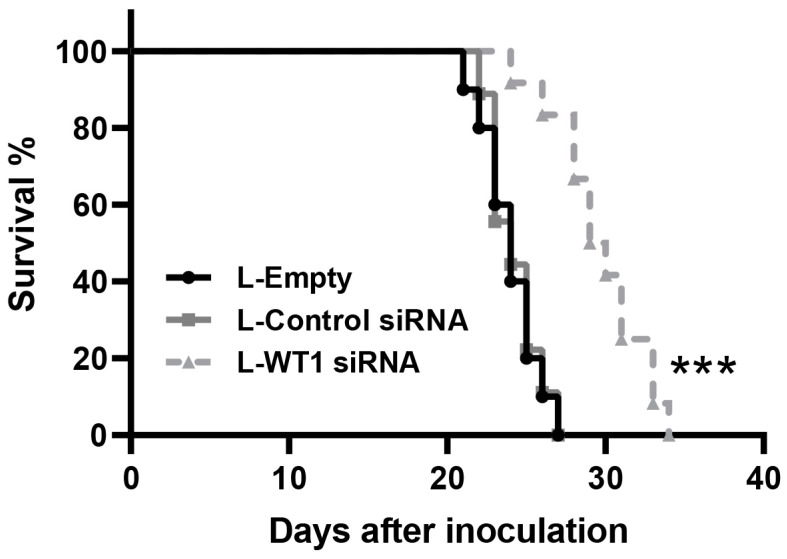
Survival assay for melanoma lung cancer metastasis model in mice treated with respiratory L-WT1 siRNA. Respiratory therapy of L-WT1 siRNA twice per week improved the survival rate of the treated animals. Log-rank test was performed as a statistical analysis. *** *p* < 0.001.

## Data Availability

All data obtained from this study are already contained in the manuscript. There is no additional dataset generated.

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
