# Peer review of "Therapeutic Effects of WT1 Silencing via Respiratory Administration of Neutral DOPC Liposomal-siRNA in a Lung Metastasis Melanoma Murine Model"

_ncrna, 2023, doi:10.3390/ncrna9020021_

Round 1

Reviewer 1 Report

The manuscript "Therapeutic effects of WT1 silencing by respiratory administration of neutral DOPC liposomal-siRNA in a lung metastasis melanoma murine mode" is well written and the experimental design is appropriate. 

Here are some comments and suggestions:

1. For figure 1C, please specify whether the results are 48 or 72 h after treatment. In the result section (line 101) it says it was 72 hr after treatment but in line 107 it specifies it was 48 h and 72 h after treatment.

2.For the results regarding figure 4A (line 167-168), were the number of metastatic foci quantified? since on the figure only the weight is shown

3. On line 171 it states the results are on Fig 3B when it should be 4B

4. In the abstract, introduction and discussion there are some mistakes. I recommend spelling and style are checked.

Author Response

Response to reviewer

The manuscript "Therapeutic effects of WT1 silencing by respiratory administration of neutral DOPC liposomal-siRNA in a lung metastasis melanoma murine mode" is well written and the experimental design is appropriate. 

Here are some comments and suggestions:

  1. For figure 1C, please specify whether the results are 48 or 72 h after treatment. In the result section (line 101) it says it was 72 hr after treatment but in line 107 it specifies it was 48 h and 72 h after treatment.

Based on this comment, the legend of the figure 1 (B and C) has been modified to match the treatment time for the mentioned experiment in each panel.

  1. For the results regarding figure 4A (line 167-168), were the number of metastatic foci quantified? since on the figure only the weight is shown.

The number of metastatic nodules cannot be quantified as they are merging with adjacent ones. Moreover, metastatic nodules are present in the whole depth of the lung tissue and, because of this, the weight of the whole lungs was considered as a quantitative measure of metastatic presence. A comparison with healthy lungs’ weight was provided as a dotted line in the graph to help to understand the extension of the tumor invasion. This explanation was added at line 170 to further clarify this measurement.

  1. On line 171 it states the results are on Fig 3B when it should be 4B

We appreciate to have noted this mistake, the proper correction was made.

  1. In the abstract, introduction, and discussion there are some mistakes. I recommend spelling and style are checked.

As suggested, the whole manuscript has been revised and English grammar has been corrected by a native English speaker colleague.

Reviewer 2 Report

In this manuscript, the authors attempted to specifically silence WT1 using L-siRNA via pulmonary administration. The idea, concept and theme of the study is attractive, however the presentation bad and the authors failed to address their study in a proper way for broader audiences.

The manuscript needs substantial revision in terms of presentation and scientific language, however the basic concept and idea can be accepted after careful revision.

Therefore, the manuscript is not suitable in its current form.

Following are some suggestions,

1-     The idea or the study conducted is useful, but the presentation is very poor. The authors used the word “being” many times without noticing that the meaning becomes changed.

2-     The manuscript needs substantial revision especially the presentation and English language. Moreover, the authors need to write the full word at first and then use abbreviation. For example, “WT1” Line 52.

3-     What is the reason behind using exact 72h transfection time? It would have been better to analyze the at multiple time point, any specific reason?

4-     The authors should have added a table mentioning the size of the L-siRNA. In fact further data about the stability and morphology would also be great.

5-     Proper style for referencing should be followed. The style is not uniform.

Author Response

Response to reviewer

In this manuscript, the authors attempted to specifically silence WT1 using L-siRNA via pulmonary administration. The idea, concept and theme of the study is attractive, however the presentation bad and the authors failed to address their study in a proper way for broader audiences.

The manuscript needs substantial revision in terms of presentation and scientific language; however the basic concept and idea can be accepted after careful revision.

Therefore, the manuscript is not suitable in its current form.

Following are some suggestions,

1. The idea or the study conducted is useful, but the presentation is very poor. The authors used the word “being” many times without noticing that the meaning becomes changed.

Thank you for letting us know the need to improve the language. We submitted the manuscript for an extensive review of the English grammar by native scientist colleagues. We hope their review of the text makes our work more suitable for an international audience.

2. The manuscript needs substantial revision especially the presentation and English language. Moreover, the authors need to write the full word at first and then use abbreviation. For example, “WT1” Line 52.

Following the reviewer’s suggestion, English grammar and scientific language was reviewed and modified. All full word’s abbreviations were carefully revised and properly added when needed.

3. What is the reason behind using exact 72h transfection time? It would have been better to analyze the at multiple time point, any specific reason?

The time between doses was chosen according to previous findings from our group, showing the maximum level of protein silencing can be achieved at 48 hours and this effect start to decrease at 96 hours after siRNA administration. For this reason, 72 hours between treatments was chosen as a time point that can maintain the maximum effect of silencing. The proper explanation was added at methodology at line 329.

https://doi.org/10.1158/0008-5472.CAN-05-0530

4. The authors should have added a table mentioning the size of the L-siRNA. In fact, further data about the stability and morphology would also be great.

An explanation about the morphology and further characteristics of our liposomal vesicle was added at lines 265-270, adding the citation to the previous work of our team that describes how the irregular flow inside the tumor’s vasculature increase the liposome’s delivery to the tumoral tissue.

5. Proper style for referencing should be followed. The style is not uniform.

Citation and references’ format have been generated following the “numbered” style on EndNote 20. Additionally, we also corrected the style of referencing when we were mentioning authors in the text.

Reviewer 3 Report

The manuscript entitled “Therapeutic effects of WT1 silencing by respiratory administration of neutral DOPC liposomal-siRNA in a lung metastasis melanoma murine model” needs very significant modification before it can be accepted for publication. Overall, the paper is poorly written and there is ample scope for authors to do additional experiments in order to defend their work. The introduction lacks adequate information about the previous study findings for readers to follow the present study rationale and procedures. My detailed comments are as follows:

Major comments:

1. Figure 1A data should be supported by qRT-PCR data. Further, transwell assay in figure 2B, C should be supported by wound healing assay

2. In figure 2B there is significant decrease of cell viability in control siRNA compared to untreated cell. Authors should explain why the cell viability decreases in control treated siRNA

3. In figure 2B the authors have claimed that cell viability decreases by 40percent were as in figure 2C the authors claimed that just 10 percent cells showed apoptosis. The authors should explain why there is such a difference between decrease in cell viability and apoptosis

4. Figure 4 the authors claimed that silencing of WT1 reduces the progression of cancer in lungs. This data should be supported by IHC images of lungs depicting the expression pattern of various cancer markers e.g. ki67 in various groups of treated mice.

Minor comments:

1. Grammatical errors should be corrected thoroughly 

2. Abstract should reflect the results and conclusion clearly

Author Response

Response to reviewer

The manuscript entitled “Therapeutic effects of WT1 silencing by respiratory administration of neutral DOPC liposomal-siRNA in a lung metastasis melanoma murine model” needs very significant modification before it can be accepted for publication. Overall, the paper is poorly written and there is ample scope for authors to do additional experiments in order to defend their work. The introduction lacks adequate information about the previous study findings for readers to follow the present study rationale and procedures. My detailed comments are as follows:

 Major comments:

  1. Figure 1A data should be supported by qRT-PCR data. Further, transwell assay in figure 2B, C should be supported by wound healing assay.

We understand the reviewer’s suggestion to include a qPCR analysis; however, we consider we have the confirmation of an effective WT1 silencing by quantifying the WT1 protein (the effector molecule) by Western Blot densitometry comparison between the groups.

We consider that migration and invasion can be better assessed and having a more conclusive result using transwell experiments, since cells require to actively migrate and even modify its diameter and shape to move, closer to the environment conditions metastatic cells find while traveling and invading other tissues.

While both recommendations are valid to support our findings, we consider we have confirmatory experiments that demonstrates the objective of showing WT1 protein silencing and the reduction on cellular motility as supported by our current data.

  1. In figure 2B there is significant decrease of cell viability in control siRNA compared to untreated cell. Authors should explain why the cell viability decreases in control treated siRNA.

Based on this comment, we have included (line 116) the explanation of this decrease in control siRNA corresponding to the cytotoxic effect of the use of the transfection agent that carries the siRNA molecules, even though this effect over viability is relatively small compared to the WT1 siRNA group.

  1. In figure 2B the authors have claimed that cell viability decreases by 40percent were as in figure 2C the authors claimed that just 10 percent cells showed apoptosis. The authors should explain why there is such a difference between decrease in cell viability and apoptosis.

An explanation of multifactorial contribution over viability was added in the corresponding paragraph in the results section (line 123). Since we are using AlamarBlue as an indirect indicator of cell’s viability, this effect is result of the contribution of apoptosis induction and arrest of the cell’s cycle.

  1. Figure 4 the authors claimed that silencing of WT1 reduces the progression of cancer in lungs. This data should be supported by IHC images of lungs depicting the expression pattern of various cancer markers e.g., ki67 in various groups of treated mice.

While having this mechanistic description of the effects of silencing WT1 over the expression of other downstream proteins would be of value, this further explanation of the involvement of different pathways that leads to tumor reduction are beyond the scope of our study. We focus on the effect of using WT1 targeted siRNA over the progression of metastases as measured by tumor size and survival. However, in a future study that we are planning as a continuation for further explaining the effect of WT1 silencing over lung metastases, we would like to explore a combined therapy with chemotherapy, and there we plan to go deeper in the study of mechanisms as well as the contribution of local inflammation and immune cell activation and recruitment.

 Minor comments:

  1. Grammatical errors should be corrected thoroughly. 

An extensive English grammar revision was performed by a native speaking scientist improving the final state of the manuscript.

  1. Abstract should reflect the results and conclusion clearly.

Abstract has been modified to show clearer the conclusions of our study.

Round 2

Reviewer 3 Report

The manuscript entitled “Therapeutic effects of WT1 silencing by respiratory administration of neutral DOPC liposomal-siRNA in a lung metastasis melanoma murine model” has been  modified as per reviewer comments. Hence, should be  accepted for publication